# Transcriptome Analysis Revealed a Positive Role of Ethephon on Chlorophyll Metabolism of *Zoysia japonica* under Cold Stress

**DOI:** 10.3390/plants11030442

**Published:** 2022-02-05

**Authors:** Jiahang Zhang, Zhiwei Zhang, Wen Liu, Lijing Li, Liebao Han, Lixin Xu, Yuhong Zhao

**Affiliations:** 1College of Grassland Science, Beijing Forestry University, Beijing 100083, China; 15104490515@163.com (J.Z.); zhangzw559@163.com (Z.Z.); bxliuwen@163.com (W.L.); llj15122428388@163.com (L.L.); hanliebao@163.com (L.H.); 2CCTEG Ecological Environment Technology Co., Ltd., Beijing 100013, China; 3Animal Science College, Tibet Agriculture & Animal Husbandry University, Nyingchi 860000, China

**Keywords:** chlorophyll content, cold stress, ethephon, transcriptome, *Zoysia japonica*

## Abstract

*Zoysia japonica* is a warm-season turfgrass with a good tolerance and minimal maintenance requirements. However, its use in Northern China is limited due to massive chlorophyll loss in early fall, which is the main factor affecting its distribution and utilization. Although ethephon treatment at specific concentrations has reportedly improved stress tolerance and extended the green period in turfgrass, the potential mechanisms underlying this effect are not clear. In this study, we evaluated and analyzed chlorophyll changes in the physiology and transcriptome of *Z. japonica* plants in response to cold stress (4 °C) with and without ethephon pretreatment. Based on the transcriptome and chlorophyll content analysis, ethephon pretreatment increased the leaf chlorophyll content under cold stress by affecting two processes: the stimulation of chlorophyll synthesis by upregulating *ZjMgCH2* and *ZjMgCH3* expression; and the suppression of chlorophyll degradation by downregulating *ZjPAO*, *ZjRCCR*, and *ZjSGR* expression. Furthermore, ethephon pretreatment increased the ratio of chlorophyll a to chlorophyll b in the leaves under cold stress, most likely by suppressing the conversion of chlorophyll a to chlorophyll b due to decreased chlorophyll b synthesis via downregulation of *ZjCAO*. Additionally, the inhibition of chlorophyll b synthesis may result in energy redistribution between photosystem II and photosystem I.

## 1. Introduction

*Zoysia japonica**(**Z. japonica**)* is a warm-season turfgrass species with good traffic stress tolerance and minimal maintenance requirements. It is widely used in China for sports turf, landscaping, and soil and water conservation [1,2]. However, in Northern China, the use of *Z. japonica* is relatively limited due to its short green period [3]. Temperature is the main factor affecting its natural distribution, turf quality, and popularization [4].

Low-temperature stress affects photosynthesis [5], the cell membrane [6,7], antioxidant systems [8], and other physiological and biochemical processes in plants [9]. Low-temperature conditions especially alter photosynthesis functions [10], among which light energy absorption, gas exchange, and carbon assimilation are the most low-temperature susceptible processes [11]. Chlorophyll (Chl) fluorescence is reduced in *Zoysia japonica* (zoysiagrass) under natural chilling stress, suggesting that photosystem (PS) II is impaired by photoinhibition [12,13]. Short-term cold stress (2 and 72 h cold treatment at 4 °C) can also induce oxidative stress and inhibit photosynthesis in *Z. japonica* [14]. However, the effect that a pretreatment with ethephon, a commercially available plant growth regulator, has on *Z. japonica* under long-term cold stress is still unclear.

Cold stress affects both the synthesis and degradation of Chl [15,16]. The decreased Chl content under low-temperature stress may be due to the inhibition of Chl biosynthesis and/or the acceleration of Chl degradation [17]. In rice, the Chl synthesis decrease under low-temperature conditions is triggered by the inhibition of δ-aminolevulinic acid (ALA) synthesis and the suppression of the protochlorophyllide (Pchlide) conversion into Chl due to the downregulation of the *protochlorophyllide oxidoreductase (POR)* gene [18]. A transcriptome study in *Z. japonica* revealed that 72 h of cold stress treatment was associated with the downregulation of several differentially expressed genes (DEGs) involved in Chl synthesis and the upregulation of DEGs involved in Chl degradation. Specifically, two genes were upregulated: the gene encoding ferrochelatase (Hem H), which catalyzes the insertion of ferrous iron into protoporphyrin IX to form protoheme that departs from the Chl biosynthetic pathway, and the gene encoding pheophorbide *a* oxygenase (PAO), which catalyzes the porphyrin macrocycle cleavage of pheophorbide a to generate a primary fluorescent Chl catabolite [14,19].

Ethylene acts as a plant hormone actively involved in plant stress response [20,21,22]. However, as a gaseous agent, the direct application of ethylene in production practice is difficult [23]. The disadvantages associated with its application can be overcome with ethephon, an ethylene-releasing reagent with great potential in outdoor use [23,24]. Interestingly, using a certain concentration of ethephon can improve the cold resistance of grape (*Cabernet Sauvignon*) [25], larch (*Larix gmelinii*) [26], and banana seedlings (*Musa*
*× paradisiaca*) [27]. At a specific concentration, ethephon can help to maintain a high Chl content in a submergence-tolerant rice (*Oryza sativa* L.) cultivar during submergence [28]. An ethephon pretreatment at a concentration of 150 mg L^−1^ can alleviate stress-related injuries in *Z. japonica* and reduce the loss of Chl under low-temperature conditions [29,30]. However, the mechanism underlying the effect of ethephon pretreatment has not been investigated and analyzed.

RNA sequencing (RNA-Seq) is a next-generation sequencing application with some clear advantages over existing sequencing methods [31,32]. Transcriptome analysis has also been applied to turfgrass species, including the bermudagrass [33], Kentucky bluegrass [34], creeping bentgrass [35], and *Z. japonica* [14]. A transcriptome analysis performed by Wang et al. [36] revealed that the gene families encoding auxin signal transduction proteins, ABA signal transduction proteins, and WRKY and bHLH transcription factors might represent the most critical components for salt-stress regulation in *Z. japonica*. Gene expression changes on the whole transcriptome level associated with ethephon pretreatment under cold stress have been rarely studied in *Z. japonica*.

The objective of this study was to explore the mechanism underlying the effect of ethephon on cold tolerance in *Z. japonica* and to prolong the green period of *Zoysia japonica* under cold stress. We assessed this effect by identifying the genes responding to ethephon treatment and analyzing the ethephon-induced key regulatory genes affecting the Chl metabolism in *Z. japonica* under cold stress.

## 2. Results

### 2.1. Chlorophyll Content in Leaves

The Chl analysis showed that CE plants had a significantly higher Chl content than CS plants (*p* < 0.05) (Figure 1A), demonstrating that ethephon pretreatment increased the Chl content under cold stress. Figure 1B shows that the ratio of Chla to Chlb did not change significantly, except for a slight increase in CE plants than in CS plants (*p* = 0.147) (Figure 1B). Moreover, the Chla/Chlb ratio change percentage relative to day 0 of CE plants was significantly lower than that of CS plants (sig = 0.027, 0.001 < 0.05) (Figure 1C), which indicated that ethephon pretreatment suppressed the decline of the Chla/Chlb ratio under cold-stressed conditions. Combined with the data in Figure 1B, it is affirmed that ethephon pretreatment could increase the Chla/Chlb ratio under cold stress.

### 2.2. Sequence Assembly

A total of 75 Gbp of clean sequencing data were obtained by a transcriptome analysis of all combined samples; the smallest clean data set had 6.7 Gbp of sequencing data and the largest had 9.86 Gbp. After removing low-quality reads, an average of 55, 554, and 958 high-quality clean reads per sample was obtained, accounting for 97.57% of the sample’s average raw reads. The percentage of Q30 bases reached over 90.51%, and the GC content of the samples varied between 49.73% and 53.48%. The clean reads of each sample were aligned with the reference genome of *Z. japonica*, reaching 93.06% of aligned reads for the sample with the highest efficiency and 88.08% of aligned reads for the sample with the lowest efficiency (Table 1). Thus, the sequencing data cleaning process generated a data set of high-quality reads that were mapped to the reference genome with a high efficiency and met the analysis requirements of subsequent tests.

### 2.3. Global Gene Expression Analysis

The comparison of CS plants with NT plants (CS vs. NT) identified a total of 16,359 DEGs in the leaves, among which 9199 were upregulated, and 7160 were downregulated (Figure 2a). Gene ontology (GO) and an enrichment analysis identified 516 biological processes, 337 molecular functions, and 128 cellular components (Appendix A). Among 16,359 cold-stress-related DEGs identified by the CS vs. NT comparison, there were 1618 DEGs with annotations in 113 KEGG pathways; 924 of those DEGs were upregulated, and 694 were downregulated (Appendix A).

The most DEGs in the leaves were identified by comparing CE plants with NT plants (CE vs. NT), with a total of 16,936 DEGs, among which 8907 were upregulated, and 8029 were downregulated (Figure 2b). GO and enrichment analysis identified 519 biological processes, 337 molecular functions, and 128 cellular components (Appendix A). Among 16,359 DEGs, 1777 were annotated in 112 KEGG pathways, which included 931 upregulated and 846 downregulated DEGs (Appendix A).

The heat map showed the overall effect of ethephon treatment under cold stress on transcription, and visualized how ethephon regulated the effect of cold stress on transcriptome (Appendix A). These results indicated that ethephon induced changes in the gene expression of *Z. japonica* under cold stress. Therefore, only the genes most relevant to ethephon application under cold stress are focused on in the study.

### 2.4. qRT-PCR Confirmation

These genes displayed a single dissociation peak and linearity between target cDNA and Ct values (Figure 3), showing that the genes used for qRT-PCR were consistent with the RNA-Seq results (Pearson’s r = 0.94, *p* < 0.001).

### 2.5. Differential Gene Expression Related to Photosynthesis-Antenna Proteins and Porphyrin and Chlorophyll Metabolism

In the CE vs. NT comparison, the most-enriched KEGG pathways related to chlorophyll included “porphyrin and Chl metabolism” (Figure 4). The results indicated that ethephon can regulate chlorophyll metabolism to improve cold tolerance. Meanwhile, in land plants, the only depots of Chlb are antenna complexes, which are composed of light harvesting complex (Lhc) proteins, [37] and Chlb content can regulate light-harvesting complexes levels in plants (Ayumi, 2019). Therefore, we analyzed the DEGs, which were derived by matching the DEGs from CE vs. NT with those from CS vs. NT ((CE vs. NT) vs. (CS vs. NT)), annotated in KEGG pathways of porphyrin and Chl metabolism (osa00860) (Appendix A) and photosynthesis-antenna proteins (osa00196) (Appendix A).

The comparison of CS vs. NT identified seventeen DEGs in the porphyrin and Chl metabolism (Figure 5c). Under cold stress, *ZjUroD1*, *ZjUroD2*, *ZjChlP1*, *ZjChlP2*, *ZjNYC1*, *ZjNADPH1*, *ZjPAO*, *ZjGSA*, *ZjPBGD*, *ZjGUS1*, *ZjHemA*, *ZjCLH*, *ZjSGR*, *ZjNADHB*, and putative *ZjCOX10* were upregulated, whereas *ZjHY2* and *ZjCPX* were downregulated (Figure 5a). Twenty-four DEGs were identified by the comparison of CE vs. NT (Figure 5c). Ethephon pretreatment under cold stress was associated with the upregulation of *ZjUroD1*, *ZjUroD2*, *ZjChlP1*, *ZjChlP2*, *ZjMgCH2*, *ZjMgCH3*, *ZjNYC1*, *ZjNADPH1*, *ZjGSA*, *ZjPBGD, ZjGUS1, ZjHemA, ZjNADHB*, and putative *ZjCOX10*, and the downregulation of *ZjNADPG2*, *ZjPAO*, *ZjCAO*, *ZjHY2*, *ZjCPX*, *ZjHMOX1*, *ZjMPE cyclase*, *ZjRCCR*, *ZjTPM*, and *ZjSGR* (Figure 5a). There were sixteen DEGs found both in CS vs. NT and CE vs. NT (Figure 5c).

Based on the CS vs. NT comparison, we identified three DEGs in the photosynthesis-antenna protein pathway (Figure 5d). Under cold stress, *ZjLhcb1* and *ZjLhcb2* were upregulated, and *ZjLhca2* was downregulated (Figure 5b). Eight DEGs were identified by the comparison of CE vs. NT (Figure 5d): *ZjLhca1, ZjLhca3, ZjLhca5, ZjLhcb1* were upregulated, and *ZjLhca2, ZjLhca4, ZjLhcb6, ZjLhcb7* were downregulated by ethephon pretreatment under cold stress (Figure 5b). There were two DEGs found in both CS vs. NT and CE vs. NT (Figure 5d).

## 3. Discussion

### 3.1. Gene Expression Analysis of Porphyrin and Chl Metabolism

The Chl content is an important physiological index of plant tolerance to cold stress [38,39]. However, the effects of cold stress on Chl metabolism are modified by ethephon treatment. We found that ethephon pretreatment elevated the Chl content by regulating genes involved in Chl synthesis, the PAO-dependent Chl degradation pathway, and the Chl cycle (Figure 6a,b).

In our DEG analysis, we identified two genes, *ZjMgCH2* and *ZjMgCH3*, encoding the magnesium (Mg)-chelatase enzyme functioning in Mg insertion into protoporphyrin IX (PIX). The Mg-chelatase acts at the branch point between the heme and Chl biosynthetic pathways [40]. Therefore, the enzyme is a crucial site for the regulation of the flow of tetrapyrrole intermediates through the Chl branch of this pathway from the common PIX intermediate [41]. Thus, the observed upregulation of *ZjMgCH2* and *ZjMgCH3* by ethephon pretreatment might have increased Chl synthesis, which eventually led to an elevated Chl content under cold stress.

*ZjMPE* cyclase encodes the Mg-protoporphyrin IX monomethyl ester (MPE) cyclase, which catalyzes the transformation of MPE into divinyl protochlorophyllide (DV-Pchlide). We found that *ZjMPE* cyclase was downregulated by ethephon pretreatment, which inhibited DV-Pchlide synthesis. The suppression of DV-Pchlide synthesis might indicate the slowdown of plant yellowing because previous research showed that DV-Pchlide is ubiquitous in the etiolated tissues of higher plants [42]. In addition, DV-Pchlide, along with other intermediates of the biosynthetic porphyrin pathway, tends to cause photo-oxidative damage in the chloroplasts after light absorption [43].

*CAO* encodes the Chla oxygenase, which catalyzes the Chlb formation by the oxygenation of Chla [44]. Ethephon pretreatment might have suppressed the conversion from Chla to Chlb under cold stress by downregulating *ZjCAO*. Therefore, the ethephon mediated an increase in the value of Chla/Chlb ratio, which is suggested to be an important determinant of maximum fluorescence intensity [45]. *CLH* encodes the chlorophyllase involved in Chl catabolism, catalyzing stress-induced Chl breakdown [46]. *ZjCLH* was upregulated under cold stress, whereas its expression level was not changed under cold stress with ethephon pretreatment. Therefore, ethephon pretreatment might have inhibited Chl degradation.

PAO and red Chl catabolite reductase (RCCR) are key enzymes involved in the PAO-dependent Chl degradation pathway [47]. The removal of phytol and Mg from Chla generates pheophorbide a. *ZjPAO* and *ZjRCCR* catalyze the conversion of pheophorbide a to red Chl catabolite, which is used to produce primary fluorescent Chl catabolites [48]. In our study, cold stress decreased Chl content by increasing *ZjPAO* gene expression in *Z. japonica* plants. In contrast, ethephon pretreatment inhibited Chl catabolism under cold stress by downregulating the gene expression of *ZjPAO* and *ZjRCCR*.

*SGR* (Stay Green) is a regulator gene in Chl degradation [49]. RNA interference in the silencing of a homolog of the rice candidate gene in *Arabidopsis* demonstrated a stay-green phenotype [50]. Under cold stress, we found that *ZjSGR* was upregulated in *Z. japonica* plants, while it was downregulated by ethephon pretreatment. Thus, the downregulation of *ZjSGR* by ethephon pretreatment might have led to an elevated Chl content in *Z. japonica* under cold stress, but the underlying mechanism remains unknown. However, ethephon treatment might have suppressed *ZjSGR* transcription because ethylene is the key regulator for *SGR* transcription [51].

### 3.2. Effect of Ethephon Application on Chl Content under Cold Stress

CE plants maintained a higher Chl content and a higher Chla/Chlb ratio than control CS plants. The elevated Chl content could be caused by increased Chl synthesis and reduced Chl degradation. Ethephon treatment increased Chl synthesis by upregulating the *ZjMgCH2* and *ZjMgCH3* genes encoding Mg-catalase. Ethephon pretreatment diminished Chl degradation under cold stress by downregulating *ZjPAO*, *ZjRCCR*, and *ZjSGR*.

An elevated Chla/Chlb ratio could be due to a reduced conversion from Chla to Chlb. The Chla/Chlb ratio, which is related to antenna size, is an indicator of functional pigment equipment and light adaptation [45]. The relatively high value of Chla/Chlb ratio is an index of stress adaptation due to a smaller light-harvesting complex, making PSII less susceptible during stress [52]. Thus, in our study, the downregulation of *ZjCAO* by ethephon elevated Chla/Chlb ratio, which might have promoted the *Z. japonica* plants to adapt to stress.

### 3.3. Gene Expression Analysis of Photosynthesis-Antenna Proteins

Photosynthetic organisms use an array of light-harvesting antenna protein complexes to efficiently harvest solar energy [53]. The Lhca proteins are associated with the light-harvesting complexes of PSI, and the Lhcb proteins are associated with those of PSII [54,55]. The genes encoding the antennas of PSI and PSII are members of the *light-harvesting complex a* (*Lhca*) and the *light-harvesting complex b* (*Lhcb*) multigene family, respectively [54]. The upregulation of *ZjLhca1*, *ZjLhca3*, and *ZjLhca5* expression and downregulation of *ZjLhcb6* and *ZjLhcb7* expression indicate a coordinated PSI-stimulation and PSII-suppression strategy adapted in ethephon-pretreated plants under cold stress.

PSII catalyzes light-induced water oxidation in oxygenic photosynthesis to convert light into chemical energy [56,57]. However, PSII is the primary target of photoinhibition [58,59]. In higher plants, PSII is composed of two moieties: the core complex, which contains all the cofactors of the electron transport chain, and the outer antenna, which increases the light-harvesting capacity of the core [60]. In higher plants, the most abundant PSII-associated light-harvesting complex (LhcII) consists of homo- and heterotrimers of Lhcb1, Lhcb2, and Lhcb3 proteins [61]. The other Chla/b-binding proteins, Lhcb4, Lhcb5, and Lhcb6, also known as CP29, CP26, and CP24, exist as monomers [62,63,64]. *ZjLhcb2* gene expression was upregulated in plants under cold stress, but its expression was not changed under cold stress with ethephon pretreatment. Furthermore, the expression of *ZjLhcb6* and *ZjLhcb7* was downregulated by ethephon pretreatment under cold stress. Lhcb2 and Lhcb6 encode LhcII components [65], but Lhcb7 encodes a minor antenna protein associated with PSII [66]. Ethephon pretreatment downregulated genes encoding antenna proteins associated with PSII to minimize its light-harvesting capacity and diminish photoinhibition under cold stress.

PSI is the most efficient light energy converter in nature [67]. The efficiency of the complex is based on its capacity to deliver the energy quickly to the reaction center, which minimizes energy loss [68]. Lhca1 and Lhca2 very rapidly transfer energy to the PSI core, but these two complexes also transfer energy to Lhca3 and Lhca4 with a similar transfer rate. Although Lhca3 and Lhca4 also have a transfer energy directed to the core, their transfer rate is slow [68]. Here, we observed the upregulation of *ZjLhca1* and *ZjLhca3* by ethephon, which improved the capacity for energy delivery. Emilie et al. analyzed a PSI-LhcI complex that was identical to the WT complex, except for the substitution of Lhca4 with Lhca5; the comparison with the WT complex indicated that the energy transfer from Lhca5 to the core was faster than that from Lhca4 [69]. Thus, in our study, the upregulation of *ZjLhca5* and the downregulation of *ZjLhca4* might have increased the efficiency of energy delivery to the PSI core, but it only required a minor change in the photosynthetic apparatus.

The coordinated PSI-stimulation and PSII-suppression strategy mediated by ethephon can be further explained by the state transition theory. The state transition is a self-regulating mechanism in which photosynthetic apparatuses balance the excitation energy distribution between PSII and PSI and improve the efficiency of light energy utilization by reversibly changing the association of LhcII with PSII and PSI [70]. During State 1, LhcII is preferentially dephosphorylated and in association with PSII. During State 2, LhcII is partially phosphorylated and preferentially leaves PSII, resulting in an association with PSI [71,72]. When PSII absorbs excessive light, LhcII moves from PSII to PSI; then, the state transition occurs to equalize the electron transport between PSII and PSI, which increases the light absorption, fluorescence intensity, and activity of PSI [73,74]. In general, plants redistribute the light energy between PSI and PSII to improve the utilization for photosynthesis and prevent damage due to excessive light exposure [75]. In our study, ethephon pretreatment most likely promoted energy redistribution between PSI and PSII in *Z. japonica* under cold stress by regulating genes encoding antenna proteins.

Evidence showed that the absence of Chlb could change the orientation of pigment molecules, which could impair energy transfer [37]. In this study, a transcriptome analysis showed that spraying ethephon inhibited the synthesis of Chlb and redistributed the energy of PSI and PSII, which also confirmed the findings of previous studies.

## 4. Materials and Methods

### 4.1. Plant Material and Growth Conditions

*Zoysia japonica Steud.* cv. *Chinese Common* plants were obtained from the Turf Research Field of the Beijing Forestry University. After transplanting, the *Z. japonica* plants were randomly placed in the greenhouse of the Beijing Forestry University. The greenhouse conditions were as follows: light intensity of 400 μmol m^−2^ s^−1^, daily light period of 14 h, day/night temperature regimen of 28/18 °C, and water 2–3 times per week after transplanting. Fifteen pots of *Z. japonica* plants were transferred and acclimated in a growth chamber (day/night temperature of 28/18 °C, light intensity of 3000 lx, light period of 14 h) for approximately one month. Plants were divided into three treatment groups (five pots per group): NT, normal day/night temperature regimen (28/18 °C) with water spray pretreatment as control; CS, cold stress (4 °C) with water spray pretreatment; and CE, cold stress (4 °C) with ethephon spray pretreatment. Ethephon pretreatment was carried out on day 1. Leaves were thoroughly sprayed with ethephon solution until liquid drops started to fall off. The concentration of ethephon was more than 150 mg/L, which was determined by our team in a pervious study [29]. Starting on day 10, the CS and CE plants were subjected to cold stress for 18 days (4 °C) (Figure 7). The cold treatment temperature was set according to Wei et al. [14]. At the end of treatment, fresh leaves of each group were collected.

### 4.2. Determination of Chl Content in Leaves

Chl content was measured according to the method described previously [76]. Fresh leaves (approximately 0.05 g per sample) were cut into small sections of approximately 5 mm and placed into centrifuge tubes filled with 8 mL of 95% ethanol. Tubes were stored in the dark for 48 h, and the absorption was measured at 665 nm, 649 nm, and 470 nm. The Chl content was calculated according to the following formula:Ca=13.95A665−6.88A649, Cb=24.96A649−7.32A665, CChl=Ca+Cb
$$\mathrm{Chla}\;\mathrm{content}=\mathrm{Ca}\times \frac{V}{W},\;\mathrm{Chlb}\;\mathrm{content}=\mathrm{Cb}\times \frac{V}{W},\;\mathrm{Chl}\;\mathrm{content}=\mathrm{CChl}\times \frac{V}{W}$$

In the formula, C_a_ refers to the concentration of Chla (mg L^−1^), C_b_ to the concentration of Chlb (mg/L), C_Chl_ to the total Chl concentration (mg L^−1^), V to the volume after extraction (L), and W to the sample weight (g).

We used a Chla/Chlb change percentage to calculate changes in Chla/Chlb ratio. Chla/Chlb change percentage were obtained by calculating the difference between the before-treatment (day 0) and after-treatment (day 28) values of each sample and dividing such a difference by before-treatment (day 0) and after-treatment (day 28). Chla/Chlb change percentage had positive and negative values, which indicated an increase or a decrease in Chla/Chlb after treatment, respectively:


$$\frac{\mathrm{Chla}}{\mathrm{Chlb}}\mathrm{ratio}\;\mathrm{change}\;\mathrm{percentage}\%=\frac{\mathrm{day}28-\mathrm{day}0}{\mathrm{day}0}\;\times 100\%$$


### 4.3. Total RNA Extraction, RNA-Seq Library Construction, and RNA-Seq

Total RNA was extracted from *Z. japonica* leaves according to the manufacturer’s instructions of the TaKaRa MiniBEST Plant RNA Extraction Kit (Takara, Japan). A NanoPhotometer instrument was used to evaluate RNA purity (OD_260 nm_/OD_280 nm_ and OD_260 nm_/OD_230 nm_ ratios), and an Agilent 2100 Bioanalyzer system was used to determined RNA integrity. The total RNA extracts were processed using magnetic beads with oligo(dT) primers to enrich the mRNA. The first cDNA strands were synthesized from short mRNA fragments, followed by the synthesis of the second strands. The new cDNA fragments were purified and subjected to end repair. The A bases were added to the 3′-ends, and the sequencing adapters were ligated to both ends of the cDNA fragments. The target size fragment fractions were recovered by agarose gel electrophoresis, and the PCR amplification was performed to complete the library preparation. Then, the effective library concentration was accurately quantified for library quality assurance. Illumina HiSeq^TM^2500 was used for second-generation sequencing after passing the library quality evaluation. We constructed a total of 9 cDNA libraries of *Z. japonica* from the four treatment groups with three biological replicates each: NT (NT1, NT2, NT3), CS (CS1, CS2, CS3), and CE (CE1, CE2, CE3). The samples were submitted for sequencing by Illumina HiSeq^TM^2500 (Novogene Co., Ltd., Beijing, China) (http://www.novogene.com, accessed on 23 March 2019).

### 4.4. RNA-Seq Data Processing and Assembly

The original image data files obtained by Illumina HiSeq^TM^ were converted into raw sequencing reads for analysis by CASAVA Base Calling [77]. Clean reads were obtained by filtering the original data and checking the sequencing error rate and GC content distribution. The clean reads were assembled based on sequence overlaps using the Trinity software [78]; the short sequence fragments were extended into longer fragments to obtain the fragment sets. The transcripts and unigene sequences were identified using the De Bruijin graph [79]. Clean reads were aligned with the reference genome of *Z. japonica* (http://zoysia.kazusa.or.jp, accessed on 2 February 2022), and their locations were mapped with the TopHat2 algorithm [80]. The HTSeq software (Baltimore, MD, USA) and union model were used to analyze the gene expression levels of the samples. The Sequence Read Archive (SRA) data were submitted to NCBI (PRJNA741873).

### 4.5. DEG Analysis

Sequencing quality assessment and gene expression volume analysis were performed based on the sequences located on the genome. We estimated the gene expression levels with the most commonly used method based on the fragment counts per kilobase of gene transcript per one million reads (FPKM) [81]. DEGs were detected by comparing the data from different samples. DEGs were subsequently subjected to cluster analysis and evaluated using DESeq2. The DEGs were verified by correcting the threshold P-value of the DEGs using the Benjamini–Hochberg multiple testing procedure for calibration to obtain the false discovery rate (FDR/Padj). Fold change refers to the ratio of the relative expression levels between two samples, which is processed by the shrinkage model of difference analysis software, and finally takes the logarithm based on 2 samples. Padj < 0.05 |log2FoldChange|>0 are chosen as the screening criteria for DEGs with biological duplication [82]. DEG with log2FoldChange value above 2 was considered to be significant.

### 4.6. Annotation Analysis

Gene Ontology (GO) is a comprehensive database describing gene function and can be divided into three parts: biological process, cellular component and molecular function. GO functional enrichment takes Padj < 0.05 as the threshold of significant enrichment. The Kyoto Encyclopedia of Genes and Genomes (KEGG) is a comprehensive database that integrates information on the genome, chemistry, and system functions with data of the respective enzymes and genes [83]. KEGG pathway enrichment of DEGs was analyzed by clusterProfiler R software. A pathway satisfying the threshold Padj < 0.05 was defined as a significantly enriched KEGG pathway for a DEG.

### 4.7. qRT-PCR Confirmation

Confirmation of RNA-Seq results by qRT-PCR was based on ten genes selected randomly from the comparison of CS vs. NT (Zjn_sc00026.1.g00600.1.am.mkhc, Zjn_sc00014.1.g01180.1.sm.mkhc, Zjn_sc00016.1.g04050.1.sm.mkhc, Zjn_sc00007.1.g00600.1.am.mkhc, Zjn_sc00086.1.g01640.1.am.mk) and CE vs. NT (Zjn_sc00133.1.g00260.1.am.mkhc, Zjn_sc00003.1.g01070.1.sm.mkhc, Zjn_sc00097.1.g01050.1.am.mk, Zjn_sc00017.1.g04240.1.sm.mkhc, Zjn_sc00016.1.g04940.1.am.mkhc). The cDNA preparations synthesized from the RNA samples were also used for qRT-PCR analysis based on ten randomly selected genes. Three technical repeats were processed for each of the three biological repeats originally performed per treatment. The primers were designed and synthesized by RuiBiotech Co., Ltd. (http://www.ruibiotech.com/, accessed on 2 February 2022) (Table 2). The instructions for the TB Green Premix Ex Taq kit (Tli RNaseH Plus) were obtained from Takara Bio Inc. (https://www.takarabiomed.com.cn/, accessed on 2 February 2022), and the qRT-PCR was performed using a Bio Rad CFX Connect Real-Time PCR Detection System (Bio-Rad Laboratories Co., Ltd., California, USA) (https://www.bio-rad.com/, accessed on 2 February 2022) with the following parameter settings: 95 °C for 30 s, followed by 40 cycles of 95 °C for 5 s and annealing/extension at 60 °C for 30 s. Single-product amplification was confirmed with a melt curve.

## 5. Conclusions

Based on the transcriptome and chlorophyll content analysis, ethephon pretreatment increased the leaf chlorophyll content under cold stress by affecting two processes: the stimulation of chlorophyll synthesis by upregulating *ZjMgCH2* and *ZjMgCH3* expression and the suppression of chlorophyll degradation by downregulating ZjPAO, *ZjRCCR*, and *ZjSGR* expression. Furthermore, ethephon pretreatment increased the ratio of chlorophyll a to chlorophyll b in the leaves under cold stress, most likely by suppressing the conversion of chlorophyll a to chlorophyll b due to decreased chlorophyll b synthesis via the downregulation of *ZjCAO*. Additionally, the inhibition of chlorophyll b synthesis may result in an energy redistribution between photosystem II and photosystem I.

## Figures and Tables

**Figure 1 plants-11-00442-f001:**
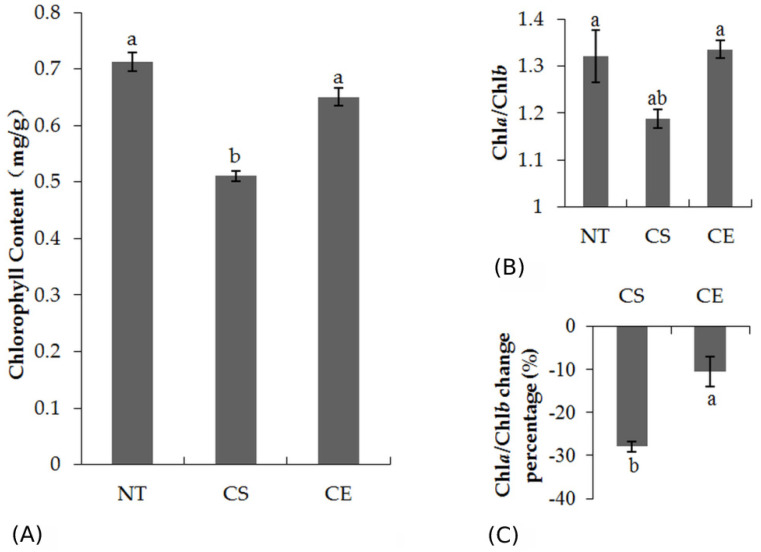
(**A**) Chlorophyll content in *Z. japonica* leaves; (**B**) Ratio of Chla to Chlb in *Z. japonica* leaves; (**C**) Chla/Chlb change percentage on day 28 relative to day 0. Error bars represent standard deviations of three independent samples. Different letters indicate significant differences at *p* < 0.05.

**Figure 2 plants-11-00442-f002:**
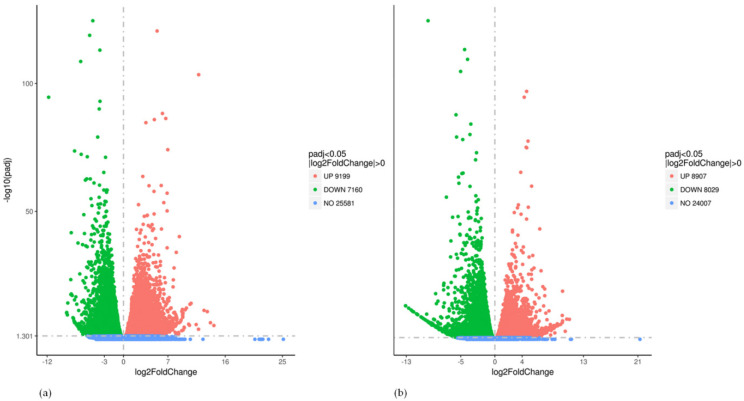
(**a**) Distribution of differential genes in volcanic map (CS vs. NT); (**b**) Distribution of differential genes in volcanic map (CEvs. NT).

**Figure 3 plants-11-00442-f003:**
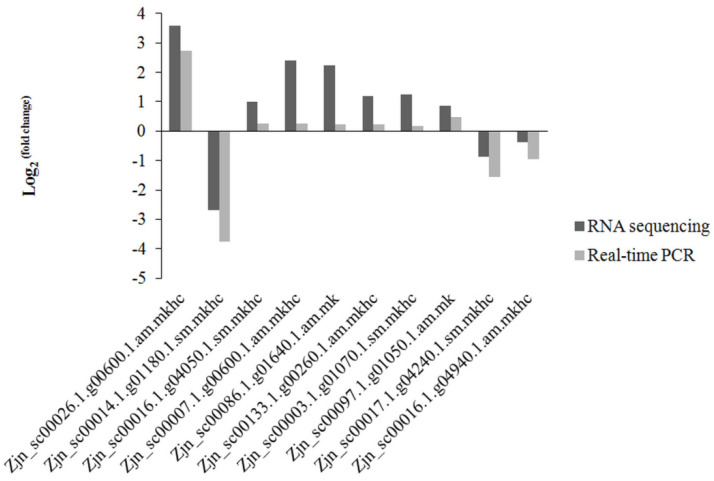
Log_2_
^(fold change)^ of genes based on RNA-Seq and qRT-PCR methods.

**Figure 4 plants-11-00442-f004:**
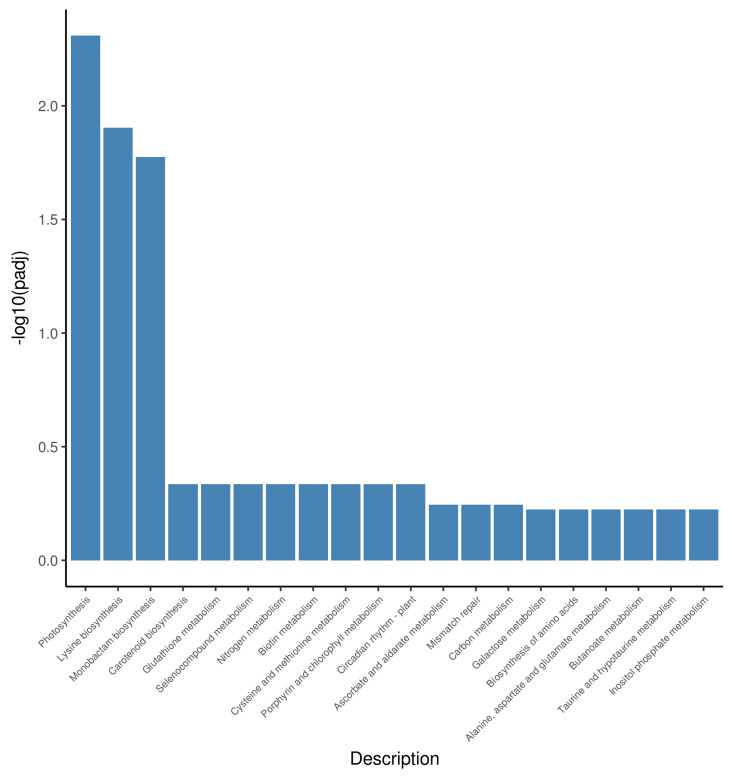
Enriched KEGG terms of CE vs. NT.

**Figure 5 plants-11-00442-f005:**
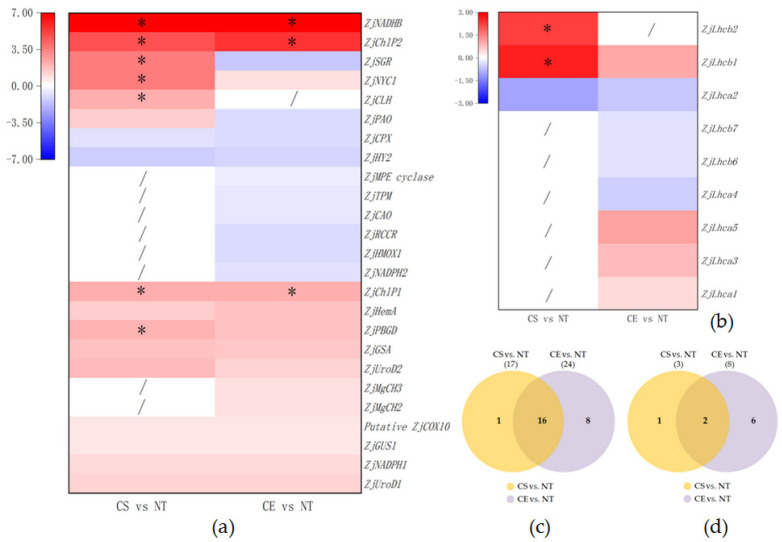
(**a**) Heat map of differentially expressed genes in porphyrin and chlorophyll metabolism pathway; (**b**) Heat map of differentially expressed genes in photosynthesis-antenna proteins pathway; (**c**) Venn diagram for all DEGs in porphyrin and chlorophyll metabolism pathway; (**d**) Venn diagram for all DEGs in photosynthesis-antenna proteins pathway; “/” indicates the DEG was not enriched in this KEGG pathway comparison; and “ * “ means the log_2_ ^(fold change)^ of this gene is greater than 2. Gene expression in log_2_ ^(fold change)^ scale was elevated with red and decreased with blue.

**Figure 6 plants-11-00442-f006:**
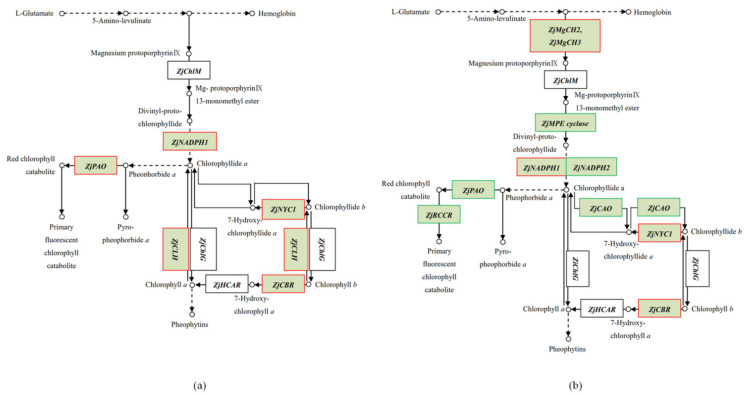
(**a**) KEGG metabolic pathway based on the comparison of CS vs. NT that matches the chlorophyll metabolic pathway; (**b**) KEGG metabolic pathway based on the comparison of CE vs. NT that matches the chlorophyll metabolic pathway. KEGG nodes indicating upregulated genes are marked in red, KEGG nodes indicating downregulated genes are marked in green. Symbols: → synthesis, ⇢ indirect synthesis, ○ synthesized product.

**Figure 7 plants-11-00442-f007:**
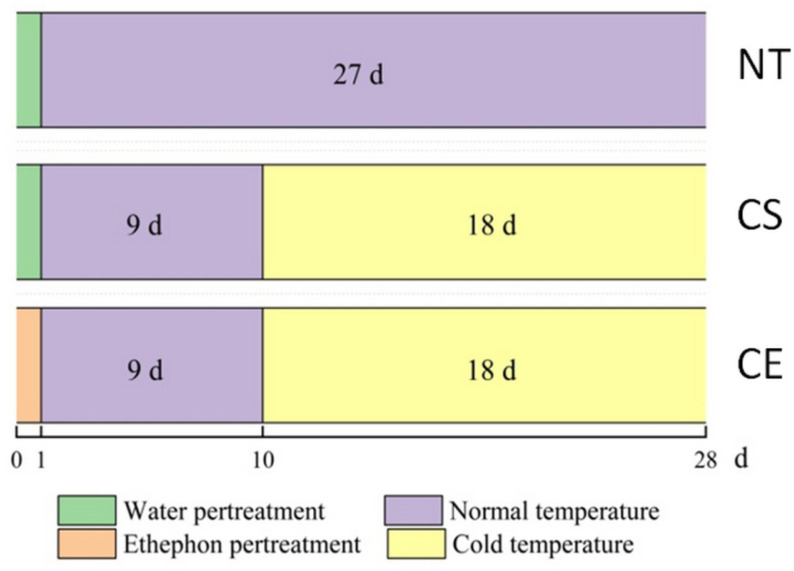
Treatment schedule for *Z. japonica* plants to test the effect of cold stress and ethephon treatment: NT (control), normal day/night temperatures (28/18 °C) with water spray pretreatment as control; CS, cold stress (4 °C) with water spray pretreatment; CE, cold stress (4 °C) with ethephon spray pretreatment.

**Table 1 plants-11-00442-t001:** An overview of the RNA-Seq data.

Sample	Raw Reads	Total Reads	Clean Reads	Clean Bases	Q30	GC Content	Total Map
				(Gbp)	(%)	(%)	(%)
CE1 ^1^	51696468	50530932	50530932	7.58	90.95	51.3	91.45
CE2	67066970	65714084	65714084	9.86	92.08	51.23	92.35
CE3	65392538	64083968	64083968	9.61	92.1	52.98	93.06
CS1 ^2^	61806546	60439330	60439330	9.07	90.91	53.48	92.11
CS2	59147316	58002528	58002528	8.7	90.51	53.46	92.19
CS3	61224860	60038194	60038194	9.01	91.67	52.75	92.79
NT1 ^3^	50204448	48478352	48478352	7.27	92.51	49.73	88.08
NT2	46169382	44677484	44677484	6.7	92.1	50.06	88.44
NT3	49753502	48029758	48029758	7.2	92.21	49.75	87.66

^1^ CE, cold stress (4 °C) with ethephon spray pretreatment. ^2^ CS, cold stress (4 °C) with water spray pretreatment. ^3^ NT, normal day/night temperature regimen (28/18 °C) with water spray pretreatment as control.

**Table 2 plants-11-00442-t002:** The target genes and primer sequences.

Gene ID	Primer Sequence
*Actin*	F:5′-GGTGTTATGGTTGGGATGG-3′
	R:5′-CAGTGAGCAGGACAGGGTG-3′
*Zjn_sc00026.1.g00600.1.am.mkhc*	F:5′-GCAGCAAGAACGAATGAT-3′
	R:5′-CTGAAGAGTGGAAGGAGAA-3′
*Zjn_sc00014.1.g01180.1.sm.mkhc*	F: GATGACAGAGATGCCAAT
	R: CGATGAATACACCAGACA
*Zjn_sc00016.1.g04050.1.sm.mkhc*	F: GGCAAGTGGTATTAGTGAA
	R: CAGTATGTGTTCCGTTGT
*Zjn_sc00007.1.g00600.1.am.mkhc*	F:GGACCTTGGACAGCATCTT
	R:CGGCGACGAAGTAGAGAAT
*Zjn_sc00086.1.g01640.1.am.mk*	F:CACGGACCAAGGACTCAAG
	R:CCAGCGTCAGTCACAAGA
*Zjn_sc00133.1.g00260.1.am.mkhc*	F:5′-GAAGGACACAGGAGTTGATG-3′
	R:5′-CCATTACCAAGGCGTCTC-3′
*Zjn_sc00003.1.g01070.1.sm.mkhc*	F:5′-ATCCTTACACCACTTCCT-3′
	R:5′-CTCATCTCGCAACACATT-3′
*Zjn_sc00097.1.g01050.1.am.mk*	F:5′-CTACCACGCTCAATCCTAT-3′
	R:5′-GTCATCCTCCTCTTCATCTT-3′
*Zjn_sc00017.1.g04240.1.sm.mkhc*	F: GGTGGTCATTGTGGATAA
	R: GGAGTCAGGTTCAGATAAG
*Zjn_sc00016.1.g04940.1.am.mkhc*	F: GCAAGAATGGAACCTGTG
	R: TCAGCAGCAATCTCATCA

## Data Availability

The data presented in this study are openly available in NCBI: reference number (PRJNA741873).

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
