# Peer review of "Transcriptome Analysis Revealed a Positive Role of Ethephon on Chlorophyll Metabolism of Zoysia japonica under Cold Stress"

_plants, 2022, doi:10.3390/plants11030442_

Round 1

Reviewer 1 Report

This work reports the effect of ethephon in Zoysia japonica under cold stress through transcriptome analysis. Samples for transcriptome analysis were collected at the end of cold treatment but the effect of ethephon in Zoysia japonica should be first evaluated by measuring content of ethanol or marker gene expression according to time progress. With this data, author can determine the best time of ethephon treatment. However, current data used the samples collected quite later after ethephon treatment and so it is not easy to explain the effect of ethephon. In addition, most of data is very descriptive and authors need to provide more biologically meaningful data. Instead of simple validation of several genes through qRT-PCR, authors need to pick up key genes from data analysis in this study, that is, genes from tables 2 and 3. Figure 6 data should be evaluated by measuring metabolites. Number of DEGs is quite a lot and authors need to more focus to more significant DEGs by considering fold change values (at least more than 4 fold change). Figures 2 and 3 are redundant data and one of them should be moved to supplemental data. Figure 4 data should be replaced by data of genes in Tables 2 and 3. KEGG enrichment analysis should adapt the fold enrichment value of selected KEGG by using observed value of the selected pathway/ the expecting value. In addition, GO enrichment analysis using p-value and fold enrichment value should be provided. 

Author Response

Point: This work reports the effect of ethephon in Zoysia japonica under cold stress through transcriptome analysis. Samples for transcriptome analysis were collected at the end of cold treatment but the effect of ethephon in Zoysia japonica should be first evaluated by measuring content of ethanol or marker gene expression according to time progress. With this data, author can determine the best time of ethephon treatment. However, current data used the samples collected quite later after ethephon treatment and so it is not easy to explain the effect of ethephon. In addition, most of data is very descriptive and authors need to provide more biologically meaningful data. Instead of simple validation of several genes through qRT-PCR, authors need to pick up key genes from data analysis in this study, that is, genes from tables 2 and 3. Figure 6 data should be evaluated by measuring metabolites. Number of DEGs is quite a lot and authors need to more focus to more significant DEGs by considering fold change values (at least more than 4 fold change). Figures 2 and 3 are redundant data and one of them should be moved to supplemental data. Figure 4 data should be replaced by data of genes in Tables 2 and 3. KEGG enrichment analysis should adapt the fold enrichment value of selected KEGG by using observed value of the selected pathway/ the expecting value. In addition, GO enrichment analysis using p-value and fold enrichment value should be provided.

Response: According to results from our previous research, different treatment time and  ethephon concentrations were evaluated by turf quality under drought conditions in various turfgrass species including Kentucky bluegrass, perennial ryegrass and zoysia grass. The treatment method used in this study can effectively alleviate the chlorophyll degradation of Zoysia japonica leaves under low temperature stress. Therefore, we used this method. Moreover, we revised the manuscript and re-draw some of the figures to improve the manuscript according to reviewer’s comments and kind suggestions. The statistical values for KEGG enrichment and GO enrichment are added in the section 4.5 and 4.6.

Reviewer 2 Report

In the presented study, the authors assessed the effect of ethephon on the metabolism of chlorophyll in Zoysia japonica exposed to cooling stress. In my opinion, the research was well designed and performed. The authors showed that ethephon protects the loss of chlorophyll in the plant by increasing the expression of genes involved in chlorophyll biosynthesis and reducing the expression of genes involved in its degradation. The advanced molecular biology techniques used allowed to fully verify the hypotheses. Before the article is accepted for publication, please add a subsection on statistical analysis.

Author Response

Point: In the presented study, the authors assessed the effect of ethephon on the metabolism of chlorophyll in Zoysia japonica exposed to cooling stress. In my opinion, the research was well designed and performed. The authors showed that ethephon protects the loss of chlorophyll in the plant by increasing the expression of genes involved in chlorophyll biosynthesis and reducing the expression of genes involved in its degradation. The advanced molecular biology techniques used allowed to fully verify the hypotheses. Before the article is accepted for publication, please add a subsection on statistical analysis.

Response: We replace some of the tables with heat maps to represent data more clearly. Statistical analysis were added as suggested.

Reviewer 3 Report

In the submitted manuscript, the authors described the transcriptomic analysis of genes which are differentially expressed in genes responding to  ethephon treatment and analyzing the ethephon-induced key regulatory genes affecting  the Chl metabolism in Z. japonica under cold stress.

The aim of the study was to clarify the mechanism underlying the effect of  ethephon on cold tolerance in Z. japonica and to prolong the green period of Zoysia japonica under cold stress.

The authors conducted extensively in silico analyses of the studied genes, combined with the experimental validation of gene expression in transcript level by quantitative real-time RT-PCR. The strong points of the manuscript is a very interesting topic and application of NGS technology. The description of identified mRNA transcripts is very interesting especially with regard to pathway connections. Below are some pointed inaccurasies:

L.33, 43,45, 80 etc…. – latin names should be written in Italic

Authors take into consideration  log2Fold Change value above 0 is rarely used, better to base on values above 1.5 or 2. Why the authors used such low values? Additionally this is in contrast to the materials and methods section 4.5 L429

Figure 4 – “Fold changes from the  first to the fifth gene were calculated based on the comparison of CS with NT (plants under cold  stress conditions were compared to plants under normal temperature conditions). Fold changes  from the sixth to the tenth gene were calculated based on the comparison of CE with NT…“ Please redraw the chart with different comparisons marked in the figure, the description itself is not appropriate

L 179 - metabolism. Under cold stress. – please rephrase the sentence

Table 2 and Table 3

Presenting the long gene name like (Zjn_sc00133.1.g00260.1.am.mkhc) - is not advisable. This name may be used for purposes of design, but it does not provide the reader with important information. My advice is to delete these weird-long names.

The tables show genes with very little or no change rate - the prefix below the table indicates this. Why mention these types of genes, they are irrelevant and appear to be overestimated. Additionally this is in contrast to the materials and methods section 4.5 L429 in which the authors reported that they take the Log2FCH genes from 2. To sum up, I evaluate the presentation of the result data on average, and the data obtained in the RNA-seq analyzes can be presented in a better way by adding analyzes such as Venn diagrams, PCA plots and others.

Author Response

Point 1: L.33, 43,45, 80 etc…. – latin names should be written in Italic.

Response 1: The format of latin names has been revised.

Point 2: Authors take into consideration log2Fold Change value above 0 is rarely used, better to base on values above 1.5 or 2. Why the authors used such low values? Additionally this is in contrast to the materials and methods section 4.5 L429.

Response 2: In this study, we thought to understand different expression level of the same DEGs in different treatment comparison is important. Therefore, we were considering to present some of the genes in the results table even though the log2Fold change value of the genes is below 2. For example, transcript of ZjLHCA1 in CE vs NT comparison was not detectable (/) but the log2Fold change is 0.49 in CS vs NT comparison. We thought this may mean something. Maybe the ethephon treatment induce the expression of ZjLHCA1 under cold stressed conditions. We revised the way to present data according to the reviewer. We transformed the table into heatmap figure. Genes with higher log2Fold change value ( above 2) are marked with " * ".

Point 3: Figure 4 – “Fold changes from the first to the fifth gene were calculated based on the comparison of CS with NT (plants under cold stress conditions were compared to plants under normal temperature conditions). Fold changes from the sixth to the tenth gene were calculated based on the comparison of CE with NT…“ Please redraw the chart with different comparisons marked in the figure, the description itself is not appropriate.

Response 3: The description of Figure 4 has been revised.

Point 4: L 179 - metabolism. Under cold stress. – please rephrase the sentence.

Response 4: Revised.

Point 5: Table 2 and Table 3 Presenting the long gene name like (Zjn_sc00133.1.g00260.1.am.mkhc) - is not advisable. This name may be used for purposes of design, but it does not provide the reader with important information. My advice is to delete these weird-long names.

Response 5: Deleted.

Point 6: The tables show genes with very little or no change rate - the prefix below the table indicates this. Why mention these types of genes, they are irrelevant and appear to be overestimated. Additionally this is in contrast to the materials and methods section 4.5 L429 in which the authors reported that they take the Log2FCH genes from 2. To sum up, I evaluate the presentation of the result data on average, and the data obtained in the RNA-seq analyzes can be presented in a better way by adding analyzes such as Venn diagrams, PCA plots and others.

Response 6: The presentation of results data has been improved by replacing tables with heatmap figures. Only those genes with the Log2FCH over 2 are marked with " * ".

Round 2

Reviewer 1 Report

Authors prepared the responses to my comments but did not address the key message, especially, validation of candidate genes from transcriptome analysis which should be selected from key pathways or biological processes emphasized in this study. In this version, authors did not perform the q-RT-PCR for candidate genes in the significant GO terms or KEGG pathways. 

DEGs should be selected by p-value and fold change together. In the table, several genes have low log2 fold changes. All other data should be reanalyzed based on p-value (at least <0.01) and log2 fold change (at least absolute value)>1). Data in figures 2 and 3 are based on only p-values.

Figure 3 need bigger boundary value than current one (-1.5, +1.5). 

GO enrichment analysis and KEGG enrichment analysis need additional criteria such as fold enrichment (ratio of observed numbers for GO term/KEGG pathway over expected numbers for GO term/KEGG pathway). With this additional criteria, authors may find out more significant GO terms and pathways. 

Physiological data should be prepared even authors used the methods in their previous papers as a supplemental data for only review purpose. Most figures need intensive improvement by addressing my comments. 

Author Response

The p-value of genes were added in the Table S5 and Table S6. We have removed the redundant data - Figure 3 into supporting materials, according to the suggestion provided in the first review report ("Figures 2 and 3 are redundant data and one of them should be moved to supplemental data."). If there is any redundant data, please give us more specific instructions. We will carefully revise it according to suggestions.